# The Indirect ELISA and Monoclonal Antibody against African Swine Fever Virus p17 Revealed Efficient Detection and Application Prospects

**DOI:** 10.3390/v15010050

**Published:** 2022-12-23

**Authors:** Liwei Li, Sina Qiao, Guoxin Li, Wu Tong, Shishan Dong, Jiachen Liu, Ziqiang Guo, Haihong Zheng, Ran Zhao, Guangzhi Tong, Yanjun Zhou, Fei Gao

**Affiliations:** 1Shanghai Veterinary Research Institute, Chinese Academy of Agricultural Sciences, Shanghai 200241, China; 2College of Veterinary Medicine, Hebei Agricultural University, Baoding 071001, China; 3Xiamen Center for Animal Disease Control and Prevention, Xiamen 361009, China; 4Jiangsu Co-Innovation Center for the Prevention and Control of Important Animal Infectious Disease and Zoonosis, Yangzhou University, Yangzhou 225009, China

**Keywords:** ASFV p17, CHO cells, epitope, indirect ELISA, recombinant PRRSV

## Abstract

Since 2018, the outbreak and prevalence of the African swine fever virus (ASFV) in China have caused huge economic losses. Less virulent ASFVs emerged in 2020, which led to difficulties and challenges for early diagnosis and control of African swine fever (ASF) in China. An effective method of monitoring ASFV antibodies and specific antibodies against ASFV to promote the development of prevention techniques are urgently needed. In the present study, ASFV p17 was successfully expressed in CHO cells using a suspension culture system. An indirect enzyme-linked immunosorbent assay (ELISA) based on purified p17 was established and optimized. The monoclonal antibody (mAb) against p17 recognized a conservative linear epitope (^3^TETSPLLSH^11^) and exhibited specific reactivity, which was conducive to the identification of recombinant porcine reproductive and respiratory syndrome virus (PRRSV) expressing p17. The ELISA method efficiently detected clinical ASFV infection and effectively monitored the antibody levels in vivo after recombinant PRRSV live vector virus expressing p17 vaccination. Overall, the determination of the conserved linear epitope of p17 would contribute to the in-depth exploration of the biological function of the ASFV antigen protein. The indirect ELISA method and mAb against ASFV p17 revealed efficient detection and promising application prospects, making them ideal for epidemiological surveillance and vaccine research on ASF.

## 1. Introduction

African swine fever (ASF) is characterized by a high fever, internal organ bleeding, and other clinical symptoms [1]. Until now, no effective vaccine or drug has been available against this disease [2,3]. ASF was first reported in August 2018 in China. In recent years, the emergence and prevalence of naturally occurring, less virulent, and naturally gene-deleted ASFV strains in domestic pigs have been identified [4,5,6,7,8]. These natural mutants showed reduced virulence and high transmissibility, causing chronic and persistent infections in pigs; however, these pathogens were continuously shed via the oral and rectal routes at a low level, leading to difficulties and challenges for early diagnosis and control of ASF in China.

Using OIE-recommended quantitative polymerase chain reaction (qPCR) and enzyme-linked immunosorbent assay (ELISA) methods, researchers can accurately judge whether pigs are infected with wild-type ASFV. Recently, a multiplex real-time qPCR was developed to provide a diagnostic tool for the differential detection of B646L, I177L, MGF505-2R, and EP402R genes [9]. For early diagnosis and the efficient prevention of circulating ASFV, antigen detection is very limited because of the marked decline in viral copy numbers. Currently, antibody detection of ASFV has become increasingly important [8]. Antibody detection methods against p30, p54, or p72 of ASFV have been the most researched and applied in clinical diagnosis [10,11,12], and it is still necessary to explore more ASFV antigens.

The ASF virus (ASFV) is a double-stranded DNA virus and is the only DNA virus transmitted by insects. ASFV contains a 170–193 kb DNA genome encoding more than 150 types of proteins [13]. Among these, p12 (*O61R*), p17 (*D117L*), p30 (*CP204L*), p54 (*E183L*), p72 (*B646L*), and CD2v are demonstrated to be involved in viral replication, immunological evasion, and the transmission of pathogens. p17 is an abundant transmembrane protein localized at the viral internal envelope, which is essential for the progression of viral membrane precursors toward icosahedral intermediates. p17 binds to the capsid protein p72 and closely connects to the inner membrane and outer shell of the ASFV virus [14]. p17 can inhibit a cGAS-STING signaling pathway to participate in complex interactions with the host for the benefit of the virus to evade the host’s defenses [15]. In particular, p17 is detected as a specific antigen in the immunoreaction of pig sera with neutralizing antibodies [16].

Mammalian cells are widely used for the expression and purification of mAb, interferons, and viral antigens, especially CHO cells [17]. CHO expressing systems have more effective and precise post-translational modifications than that in the prokaryotic system; moreover, they express recombinant proteins that are very close to natural proteins in terms of the molecular structure, physicochemical properties, and biological functions. Using a serum-free and composition-simple medium for the suspension culture contributes to massive purification and production and also increases the expression level of recombinant proteins. Herein, we expressed recombinant p17 in CHO cells using the suspension culture system and established an indirect ELISA and mAb against p17, which revealed efficient detection and offered promising application perspectives.

## 2. Materials and Methods

### 2.1. Cells, Viruses, and Sera

Various cell cultures, including 293T, SP2/0, MARC-145, porcine alveolar macrophages (PAMs), and suspension-cultured CHO cells, were maintained in our laboratory. A full-length cDNA clone of the HP-PRRSV attenuated strain vHuN4-F112 obtained by serial cell passage (GenBank accession no. EF635006) was used as the backbone for the insertion of *D117L* from SY18 (GenBank accession no. MH766894.1). The resultant recombinant virus was compared with the parental virus, vHuN4-F112. PRRSV titers in MARC-145 cells were determined using the standard median tissue culture infective dose (TCID_50_) following the Reed and Muench method [18]. Swine serum samples of a virulent ASFV strain (wild-type ASFV) were stored until further use. Swine serum samples (*n* = 155) were collected from pig farms. Serum samples positive for PRRSV, classical swine fever virus (CSFV), foot and mouth disease virus (FMDV), porcine epidemic diarrhea virus (PEDV), type 2 porcine circovirus (PCV2), and pseudorabies virus (PRV), respectively, were conserved in our laboratory.

### 2.2. Expression and Purification of Recombinant p17

Based on *D117L* sequence of the ASFV SY18 strain, pcDNA3.1-*D117L*-strep plasmid was constructed using the pcDNA™3.1 (+) vector (Thermo Fisher Scientific, Waltham, MA, USA). The recombinant eukaryotic expression vector pcDNA3.1-*D117L*-strep was transiently transfected into CHO cells and maintained in a shaking incubator at 37 °C and 8% CO_2_ at 125 r/min. The cells were supplemented with enhancers and auxiliaries after 18–22 h and incubated in a shaking incubator for 5 days. Cells were identified 1, 3, and 5 days post-transfection (dpt). Cells were collected at 5 dpt, and the recombinant p17 was purified using StrepTrap beads according to the instructions provided by the manufacturer (General Electric Company, Boston, MA, USA). The collected samples were identified by SDS-PAGE and Western blotting (WB), using an anti-strep tag antibody (1:4000, ab180957, Abcam, Cambridge, MA, USA).

### 2.3. Establishment of an Indirect ELISA against p17

The ELISA was optimized using the square titration method. The ELISA plates were first coated with different concentrations of p17 (100, 200, 400, 800, and 1000 ng/well), and ASFV-positive and negative sera were diluted from 1:50 to 1:400 to determine the optimal concentration of protein coating and serum dilution, respectively. The coating solution (carbonate or phosphate), sealing solution (5% skim milk or 5% BSA), and sealing time (37 °C for 1 h or 37 °C for 2 h) were optimized. The coating temperature (37 °C for 1 h, 37 °C for 2 h, 37 °C for 4 h, or 4 °C overnight) and secondary antibody dilution (6 × 10^3^, 1 × 10^4^, 2 × 10^4^, or 4 × 10^4^) were then optimized. Then, the optimal duration (20, 30, 45, or 60 min) of the serum and secondary antibodies was identified according to the conditions determined above. The color development conditions were optimized using TMB (5, 10, or 15 min at room temperature or 37 °C, respectively).

### 2.4. Generation and Screening of the mAb against p17

Five 4-week-old female BALB/c mice were immunized with purified p17 via intramuscular, subcutaneous, and intraperitoneal multipoint injections. The purified protein (50 µg) was mixed with MnJ(β) colloidal manganese adjuvant. A second immunization was performed 10 days after the first one, followed by a third and booster immunization, in 7 days gaps using the same method and dose. The immune response of the mice was tested by indirect ELISA, and the spleen cells of mice and SP2/0 cells were hybridized to produce hybridoma cells that secreted specific antibodies against p17. Hybridoma cells were screened by indirect ELISA, and wells with high positive values were subcloned three times using the limited dilution method. Monoclonal cells that stably secreted the specific antibody against p17 were finally selected for expanded culture and preparation of ascites. The ascites was named 6E3, purified using a Pierce™ antibody clean-up kit (Thermo Fisher Scientific), and stored at −30 °C.

### 2.5. Specific Detection and Subtype Identification of the mAb against p17

To verify the specificity of 6E3, 293T cells were transfected using pCDNA3.1, and pCDNA3.1-*D117L*-strep plasmids for indirect immunofluorescence assay (IFA) and WB analysis using mAb against p17 (6E3), or anti-strep tag antibody mentioned above. The fluorescence was visualized using an inverted fluorescence microscope (Olympus Corporation, Tokyo, Japan). The subtypes of 6E3 were identified using a monoclonal antibody isotype identification kit (Proteintech Group, Inc., Chicago, IL, USA) according to the manufacturer’s instructions. The purified 6E3 was then diluted multiplicatively (1:1000 to 1:1,024,000), and the antibody titer was measured using the indirect ELISA method established above.

### 2.6. Mapping the B-Cell Epitope

The minimal epitope recognized by the mAb was identified by WB and indirect ELISA. A truncated fragment of *D117L* gene was ligated into the prokaryotic expression vector pCold-TF and expressed in BL21(DE3) using IPTG (1 mM). The truncated protein recognized by the mAb was verified by WB. Based on these results, the p17 mutant was further truncated. Primers used in this study were listed in Table 1. Finally, the peptides were synthesized and coated onto the ELISA plates. The OD_450_ value of each short peptide recognized by the mAb was determined by indirect ELISA, and the smallest B-epitope was determined.

### 2.7. Virus Rescue and In Vitro Analysis of Virological Characteristics

The recombinant full-length cDNA clones (pA-p17 harboring the ASFV *D117L* gene) were successfully assembled using the same strategies as previously described [19]. The parental plasmid pHuN4-F112 and recombinant plasmid pA-p17 were linearized with *Swa* I, which was immediately downstream of the poly (A) tail, and then gel-purified using the QIAgen PCR purification kit (QIAgen, Hilden, Germany). Subsequently, linearized templates were then subjected to in vitro transcription using the T7 mMESSAGE mMachine kit (Ambion-Thermo Fisher Scientific, Waltham, MA, USA). MARC-145 cells were transfected with 2 μg of in vitro transcripts and 2 μL of DMRIE-C (Invitrogen). Cells were monitored daily for cytopathic effects (CPEs). Viral supernatants were collected when 80% of cells developed CPEs. The rescued viruses (P1) were passaged 20 times in MARC-145 cells by plaque purification. Viruses were harvested and designated as vA-ASFV-p17. The recombinant PRRSVs (P5, P10, P15, and P20 viral stocks) were stored at −80 °C until use. The viral plaque assay and multistep growth curves were performed as previously described [19].

### 2.8. Application of the Indirect ELISA against ASFV p17

#### 2.8.1. Coincidence Rate Experiments of Clinical Samples

Swine serum samples (*n* = 155) were analyzed using a commercial ELISA kit (ASFS-5P, Innovative Diagnostics, Grabels, France) and the established ELISA method, with optimized conditions to determine the effectiveness and coincidence.

#### 2.8.2. Specificity Testing

Serum samples of CSFV, PRV, PRRSV, PEDV, FMDV, PCV2, and ASFV were tested to determine the specificity of the ELISA method.

#### 2.8.3. Sensitivity Testing

The ASFV-positive sera were diluted at six dilutions of 1:40, 1:80, 1:160, 1:320, 1:640, and 1:1280, and were determined by the ELISA method under optimized conditions to evaluate the sensitivity.

#### 2.8.4. Detection of Specific Humoral Immunity after vA-ASFV-p17 Vaccination

For immune efficacy experiments, fifteen 30-day-old PRRSV- and ASFV-free piglets were selected and divided into three groups to evaluate the specific humoral immunity induced by vA-ASFV-p17. Each group contained five piglets that were fed separately. A 2 mL dose of vA-ASFV- p17 or vHuN4-F112 (dosage:10^5.0^ TCID_50_) was used to vaccinate each piglet (PRRSV S/P < 0.4; *n* = 5) through intramuscular cervical injections. Each piglet in the mock group was injected with 2 mL of DMEM. The PRRSV-specific antibody titers in the serum samples collected at specified time points were tested using commercial ELISA kits (IDEXX Laboratories, Westbrook, ME, USA, No. 06-40959-04), as previously described. The ASFV p17-specific antibody titers in the serum samples collected at the specified time points were tested using the indirect ELISA established above. Based on these values, the humoral immunity levels were plotted using GraphPad Prism 6.0.

### 2.9. Statistical Analysis

All experiments included at least three independent repeats. Statistical significance was analyzed using the *t*-test. Statistical significance was considered at *p* < 0.05.

## 3. Results

### 3.1. The Recombinant p17 Was Obtained Successfully from Suspension Cultured CHO Cells

The gene fragment of *D117L* was amplified (Figure 1A) and verified to be consistent with the reported gene sequence of the ASFV SY18 strain by sequencing. SDS-PAGE showed that p17 was expressed successfully and that the expression levels gradually increased with increasing transfection time. The expression levels peaked at 5 dpt (Figure 1B). Using an anti-strep tag antibody that formed clear bands of approximately 17 kDa, the purified protein was specifically detected, which was consistent with expectations (Figure 1C). Using an ASFV-positive serum as the primary antibody, the purified protein was also specifically detected (Figure 1D). These results indicated that the recombinant p17 could be successfully obtained from suspension-cultured CHO cells and specifically recognized by ASFV-positive serum as a suitable antigen for subsequent tests.

### 3.2. The Indirect ELISA against p17 Was Established and Optimized

The obtained p17 was used as the antigen in an indirect ELISA. Using the square titration method, the conditions were optimized as follows: The optimal concentration of p17 was 400 ng/well. The optimal serum dilution of ASFV-positive and ASFV-negative sera was 1:100. The coating consisted of a phosphate solution at 4 °C overnight. The sealing solution consisted of 5% BSA at 37 °C for 1 h. The secondary antibody dilution was 1:20,000. Then, the optimal durations of serum and secondary antibodies were 45 min and 45 min, respectively. The color development conditions for TMB were 15 min at 37 °C. All raw data from the optimization assays will be made available upon request.

### 3.3. The mAb against p17 Exhibited Specific Reactivity

Using the above method, the supernatant of hybridoma cells after cloning four times by limiting the dilution was determined to be positive, and the positive cells were injected into pristane-treated BALB/c mice to obtain abundant ascetic fluid. The mAb subtypes were identified as IgG2b for the heavy chain and kappa chain for the light chain (Figure 2A). The collected ascites were then purified and named 6E3. The detected antibody titer of 6E3 was 1:256,000 as per ELISA (Figure 2B). IFA showed that the 6E3 and the anti-strep tag antibody specifically recognized the expressed-p17 in CHO cells with green and red fluorescence, respectively, whereas the control did not show any fluorescence (Figure 2C). WB showed that the 6E3 recognized a specific and clear band with a molecular weight of approximately 17 kDa, whereas it did not react with the control (Figure 2D), indicating that the mAb exhibited specific reactivity against p17.

### 3.4. The mAb against p17 Recognized Specific Linear B-Cell Epitope

The *D117L* gene (1-354bp) was first divided into two segments (P1:1–60aa; P2:50–117aa) and the expression plasmids were constructed (Figure 3A). WB showed that the 6E3 recognized the P1 region only (Figure 3B). Further, the P1 fragment was truncated into three segments (P1-1: 1–25aa; P1-2: 15–42aa; P1-3: 32–60aa) and the 6E3 only recognized the P1-1 region (Figure 3C), indicating the epitope located within the P1-1 region. The primers used in the experiments are listed in Table 1.

For further precise identification of the epitope, twelve different truncated peptides were synthesized and used to coat the ELISA plates. The results showed that the 6E3 recognized the minimum epitope located at amino acids 3–11 of p17, with the sequence ^3^TETSPLLSH^11^ (Figure 3D). 

### 3.5. The Epitope Recognized by the 6E3 Was Conservative among Different Strains

To analyze the conservation of the epitope sequence, 24 representative ASFV strains from different genotypes were collected from the GenBank database (Table 2) and were aligned using MEGA. The results of percent identity and divergence showed that the consistency of *D117L* sequences among genotype I and II strains was higher than 97.2% but slightly lower than that detected in the other genotypes (Figure 4A). The alignment of amino acid sequences showed that the epitope (^3^TETSPLLSH^11^) was 100% conserved in the 24 representative strains (Figure 4B). The prediction of the p17 structure by PyMOL revealed that the epitope (^3^TETSPLLSH^11^) was located at the N-terminal of p17 (pink) and was exposed to the surface of the molecule (Figure 4C). The results above indicated that ^3^TETSPLLSH^11^ was a conserved epitope of p17 among the representative ASFV strains from different genotypes.

### 3.6. The 6E3 Specifically Recognized the Recombinant PRRSV Expressing p17

The recombinant PRRSV virus expressing the ASFV p17 was constructed as shown in Figure 5A. To evaluate the in vitro growth characteristics of vA-ASFV-p17 and vHuN4-F112, CPE, plaque morphology, and virus titers were monitored, as described previously [19]. The results showed that the CPEs of vA-ASFV-p17 developed simultaneously with those of the parental strain vHuN4-F112 (Figure 5B); vA-ASFV-p17 and vHuN4-F112 appeared similar in respect to plaque morphology and viral growth analysis (Figure 5C–E). We evaluated the foreign gene expression of vA-ASFV-p17. WB showed that the 6E3 specifically recognized the vA-ASFV-p17 producing a specific band and did not react with vHuN4-F112. As a control, a laboratory-preserved antibody against a PRRSV nonstructural protein 10 (Nsp10) was used as the primary antibody, which reacted with both viruses to produce specific bands (Figure 5F). MARC-145 cells were infected and tested for IFA by using 6E3 as the primary antibody. The results showed that the 6E3 specifically bound to the vA-ASFV- p17-infected MARC-145 cells, producing green fluorescence, whereas it reacted with the vHuN4-F112-infected MARC-145 cells, producing no fluorescence. As a control, the PRRSV N protein antibody was used as the primary antibody and reacted with both viruses to produce a specific red fluorescence (Figure 5G). PAMs were used to analyze the expression of p17 in the target cells of a PRRSV infection. The results showed that p17 specifically bound to vA-ASFV-p17-infected PAMs, producing red fluorescence (Figure 5H). By the serial cell passage of recombinant virus in vitro, IFA for vA-ASFV-p17 (P5, P10, P15 and P20 viral stocks) was performed. The results showed that the 6E3 specifically bound to vA-ASFV-p17-infected MARC-145 cells that produced red fluorescence (Figure 5I), indicating that p17 could be stably expressed in recombinant PRRSVs for at least 20 passages. These results indicated that 6E3 exhibited a specific response and could be used for the identification and detection of the recombinant virus expressing ASFV p17.

### 3.7. The ELISA Method Revealed Efficient Detection and Application Prospects

ASFV-positive serum samples at different dilutions were detected, and the results appeared positive at 1:1280 (Figure 6A), indicating the high sensitivity of the method. Serum samples of CSFV, PRV, PRRSV, PEDV, FMDV, PCV2, and ASFV were tested using the ELISA method. The S/N values of the standard ASFV-positive serum samples were significantly greater than 1.5, while the S/N values of the remaining serum samples were all less than 0.2, which met the criteria for a negative control, indicating good specificity of the method (Figure 6B). Furthermore, 50 out of 155 clinical serum samples tested positive by the established ELISA method, which coincided with the result of the commercial kit (Table 3), indicating that the established ELISA method is suitable for the diagnosis of clinical samples. Finally, ELISA was used to monitor the antibody levels in vivo of the recombinant PRRSV virus expressing ASFV p17. The groups vaccinated with vA-ASFV-p17 or vHuN4-F112 produced PRRSV-specific immune effects. All piglets showed seroconversion by 14 days post-vaccination (dpv), and the average S/P showed a peak of approximately 2.8 at 42 dpv (Figure 6C), demonstrating that vA-ASFV-p17 vaccination induced high levels of PRRSV-specific antibodies. Whereas the ASFV-specific antibody levels began to increase at 21 dpv in the piglets vaccinated with vA-ASFV-p17 and peaked at 42 dpv (Figure 6D). During the study period, the PRRSV-specific and ASFV-specific antibodies in the mock group of piglets remained negative (Figure 6C-D). In summary, the established ELISA method has great potential for the specific and effective detection of clinical ASFV infections and the antibody level in vivo of recombinant PRRSV live vector virus expressing ASFV antigen protein. 

## 4. Discussion

The ASFV has huge particles and numerous encoded proteins [1,13]. At present, most of the research on protein function remains largely unknown, but a few of the existing studies on proteins have investigated the function of p17 [14,20]. Here, we explored whether p17 could be employed as a potentially useful serological diagnostic antigen. The indirect ELISA against p17 revealed high sensitivity and good specificity and showed great potential for ASF epidemic monitoring and control (Table 3 and Figure 6). The mAb prepared from the recombinant p17 vaccination showed specific reactivity to cells transfected with a *p17*-expression plasmid and recombinant PRRSV expressing p17 (Figure 2 and Figure 5), indicating the significance and effectiveness of our tool in research on p17 function and live vector vaccines.

The CHO expressing system has more effective and precise post-translational modifications than that in the prokaryotic system. The recombinant protein obtained from this system can fold correctly to maintain its spatial conformation and protein activity. Contrastingly, the recombinant protein, produced using a prokaryotic expression system and structurally differing from the natural viral protein, primarily displays a linear epitope. CHO cells represent the most frequently applied host cell system for the industrial manufacturing of recombinant protein therapeutics [17,21]. Utilizing the suspension culture system can further increase the expression level due to the significantly increased cell density. Moreover, the suspension culture system yields relatively pure proteins owing to the serum-free culture medium and the simple purification process. Moreover, viral envelope proteins, which are difficult to express in prokaryotic systems, perform better in eukaryotic systems. In the present study, p17 was successfully expressed and purified from suspension-cultured CHO cells (Figure 1), which showed good immunogenicity in further experiments. 

There have rarely been reports on the epitopes of p17. Further investigation is needed as a possible diagnostic tool. Our results showed that the 6E3 specifically recognized the minimal linear epitope, ^3^TETSPLLSH^11^ (Figure 3). Further analysis showed that the sequence of this epitope was conserved among the 24 representative ASFV strains (Figure 4). So far, 24 ASFV genotypes have been identified based on the 3′-end sequence of the *B646L* gene, which encodes the major capsid protein p72 [22]. In 2018, the Georgia-07-like genotype II ASFV emerged in China and spread to other Asian countries [4]. We selected multiple representative strains of different genotypes to analyze the conservation of p17 and found that the epitope (^3^TETSPLLSH^11^) was 100% conserved. A further prediction of the p17 structure showed that the epitope was located in the N-terminus of p17 and was exposed to the surface of the molecule, contributing to the understanding of the ASFV protein.

ASF vaccination approaches include inactivated vaccines, subunits, DNA, live attenuated vaccines, and virus-vectored vaccines [23,24]. Several gene-deleted ASFV vaccine candidates have been reported in many countries [25,26,27,28,29]. Thus, live attenuated vaccines pose a slight risk, as in rare cases attenuated strains may regain pathogenicity, causing the spread of the disease; they also have the potential to cause post-vaccination reactions and side effects. The structural proteins p30, p54, p72, and hemagglutinin CD2v have traditionally been the main targets of subunit and DNA vaccines. More recent studies using antigen cocktails of up to 47 different ASFV genes delivered by adenovirus, alphavirus, and vaccinia virus vectors demonstrated the induction of strong antigen-specific cellular responses [2]. Although, biological safety prevention and control measures can only solve the ASF epidemic to some extent. The costs have increased greatly, which is not conducive to the healthy development of the swine industry and will also bring a series of problems such as environmental pollution and rising pig prices. The biological characteristics of the pathogen ASFV are complicated. Meanwhile, the ASFV invasion and the induction of the immune protection response have not been clearly addressed. Cellular immunity plays a vital role in the process of resisting ASFV infection.

The advantage of a live vector vaccine is that the antigen can be continuously expressed in vivo, and the vector virus itself can stimulate cellular immunity. After expressing the antigen protein of ASFV, it can effectively stimulate ASFV-specific humoral immunity and cellular immunity, thus significantly improving the immune protection effect. Considering the perspectives of biological safety and immune efficacy, live vector vaccines should be the direction of the ASF vaccine in the future. The recombinant virus rPRRSV-E2, which expressed the E2 protein of the classical swine fever virus with the attenuated PRRSV vaccine strain (vHuN4-F112) as a live vector, could provide complete immune protection for piglets to resist the challenges of CSFV and HP-PRRSV [19,30,31]. As a single-stranded RNA virus, the PRRSV attenuated vaccine strain can express foreign proteins in pigs and induce a strong immune response. PRRSV and ASFV also share a common target cell—PAMs. In this study, p17 of ASFV was selected as the foreign antigen and was inserted into the live vector of PRRSV. The recombinant virus was successfully constructed and rescued, and the immunogenicity of the recombinant virus was verified. The mAb established in this study can be a useful tool to evaluate the expression level of p17 in the recombinant virus (Figure 5), and the indirect ELISA method has great potential for the specific and effective detection of the antibody level in vivo of the recombinant virus (Figure 6), making them meaningful for future vaccine research on ASF.

Conclusively, the recombinant ASFV p17 was successfully expressed and purified from CHO suspension-cultured cells. The mAb 6E3 specifically recognized a transiently expressed p17, as well as specifically recognized recombinant PRRSV-infected cells expressing ASFV p17. These results indicate that the anti-p17 mAb generated in this study has strong specificity and the potential to be used in both basic and applied research on ASFV. Furthermore, the indirect ELISA method against p17, exhibiting good specificity and sensitivity, can potentially detect clinical ASFV infections and the antibody level of recombinant PRRSV live vector virus expressing ASFV antigen protein in vivo.

## Figures and Tables

**Figure 1 viruses-15-00050-f001:**
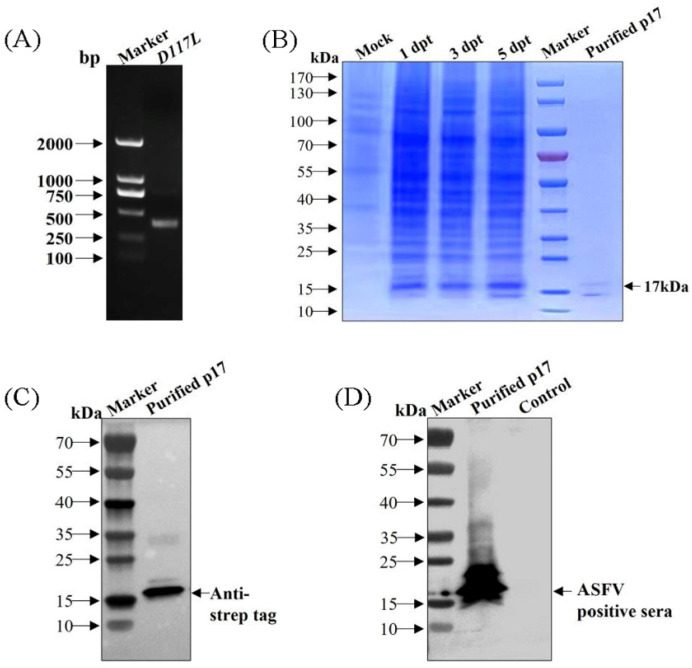
The recombinant p17 was successfully expressed and purified from CHO cells. (**A**) PCR amplification of D117L gene. (**B**) Identification of p17 expression and purification in CHO cells by SDS-PAGE. (**C**) Identification of purified p17 by WB using anti-strep tag antibody. (**D**) Identification of purified p17 by WB using an ASFV-positive serum as primary antibody.

**Figure 2 viruses-15-00050-f002:**
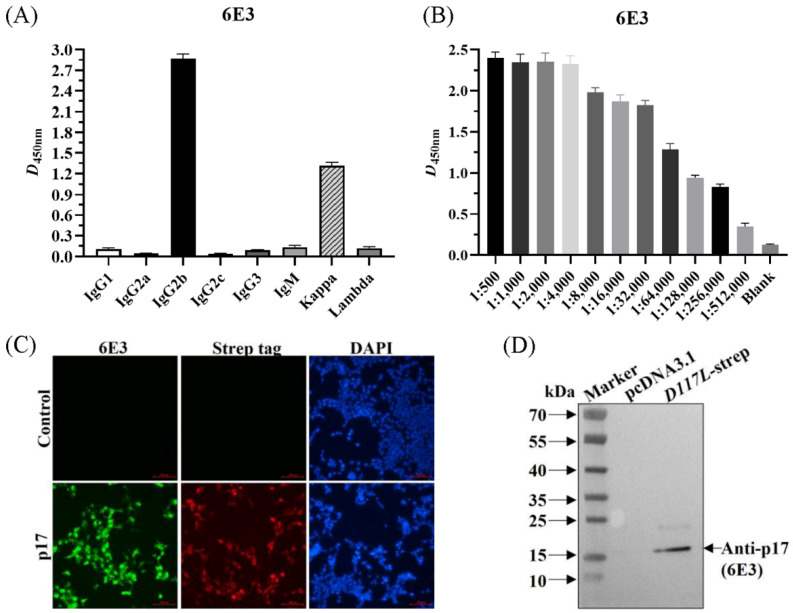
The anti-p17 mAb (6E3) specifically recognized CHO suspension cells transiently transfected with p17-expressing plasmid. (**A**) Identification of the mAb subtypes. (**B**) Identification of antibody titer of the purified 6E3 by the ELISA method. (**C**) 293T cells were transfected with pcDNA3.1-*D117L*-strep or control plasmid. Cells were fixed at 24 h post-transfection and immunostained with 6E3 as primary antibody and FITC-conjugated goat anti-mouse IgG as second antibody. Cellular nuclei were counterstained with 1 μg/mL of 4′,6′-diamidino-2-phenylindole (DAPI). (**D**) WB was conducted as treated in (**C**) to show the reactivity of 6E3.

**Figure 3 viruses-15-00050-f003:**
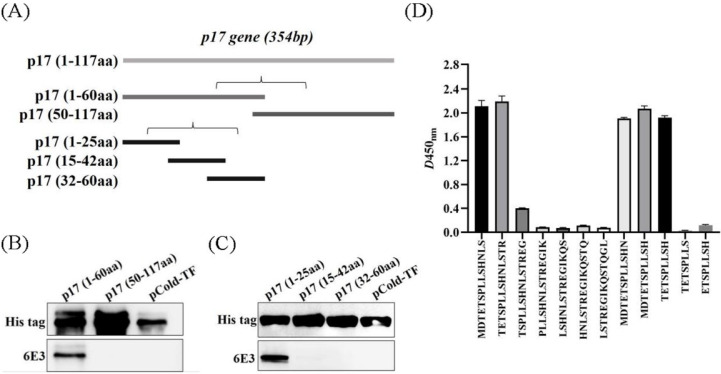
6E3 recognized specific linear B-cell epitope ^3^TETSPLLSH^11^. (**A**) Schematic diagram of D117L-truncated fragments. (**B**,**C**) A series of D117L-truncated fragments were constructed to pCold-TF and successfully expressed in E. coli BL21 (DE3) cells. 6E3 was used to detect the truncated p17 by WB using anti-His tag antibody or 6E3 as primary antibody, respectively. (**D**) Twelve different truncated peptides were synthesized and tested by ELISA to show the minimum epitope recognized by 6E3.

**Figure 4 viruses-15-00050-f004:**
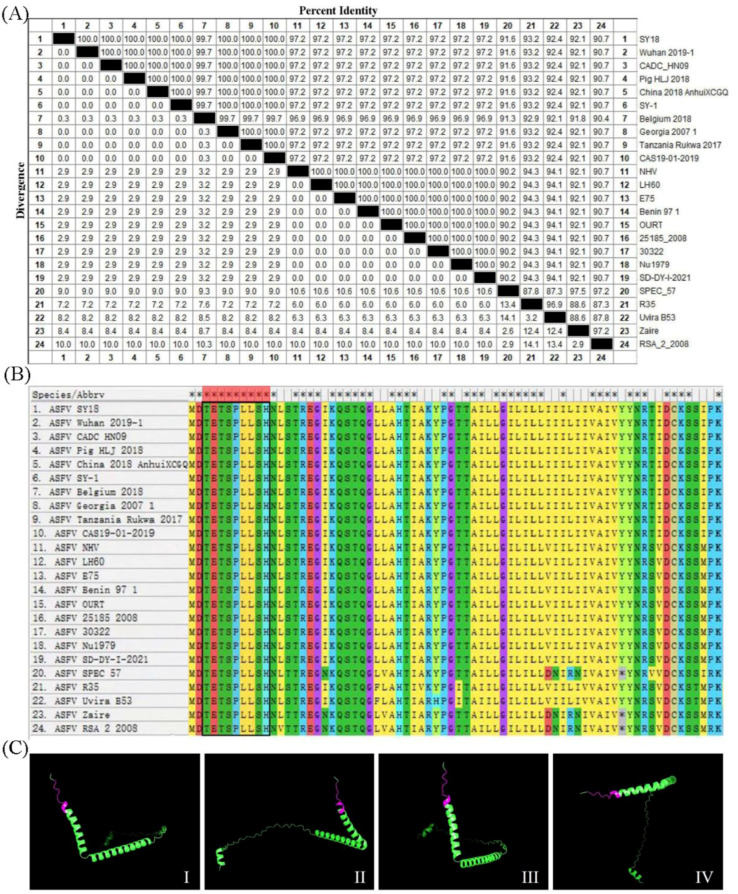
The epitope recognized by 6E3 was conservative among different strains. (**A**) The percent identity and divergence of p17 among the 24 representative ASFV strains were analyzed using MEGA. (**B**) Alignment analysis of the epitope (^3^TETSPLLSH^11^) in 24 representative ASFV strains. (**C**) Prediction of the p17 structure using PyMOL. The epitope recognized by 6E3 is displayed in pink color.

**Figure 5 viruses-15-00050-f005:**
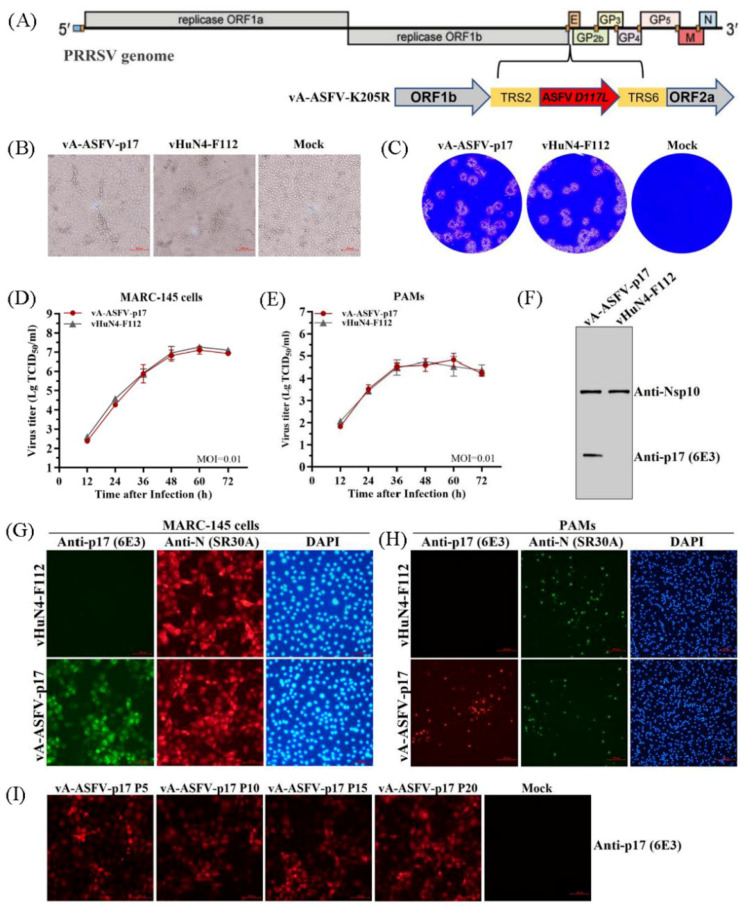
6E3 specifically recognized the recombinant PRRSV expressing p17. (**A**) The schematic representation of recombinant PRRSV virus expressing ASFV p17 construction. (**B**,**C**) CPE and plaque morphology investigation. MARC-145 cells were infected with vA-ASFV-p17 and vHuN4-F112 (MOI = 0.001). The mock control represented non-infected MARC-145 cells. MARC-145 cells were monitored or stained with crystal violet at 3 days post-infection. (**D**,**E**) Growth characteristics of vA-ASFV-p17 and vHuN4-F112 were evaluated in MARC-145 cells (**D**) and PAMs (**E**). (**F**) MARC-145 cells were infected by vA-ASFV-p17 and vHuN4-F112. WB analysis of cell lysates using 6E3 and an anti-Nsp10 antibody. (**G**,**H**) IFA against PRRSV N protein or ASFV p17 in MARC-145 cells (**G**) and PAMs (**H**) at 36 hpi with vA-ASFV-p17 and vHuN4-F112 (MOI = 0.1). Cellular nuclei were counterstained with DAPI. Scale bar = 100 µm. (**I**) Detection of ASFV p17 expression in the recombinant PRRSVs (P5, P10, P15, and P20 viral stocks) using 6E3 as primary antibody by IFA.

**Figure 6 viruses-15-00050-f006:**
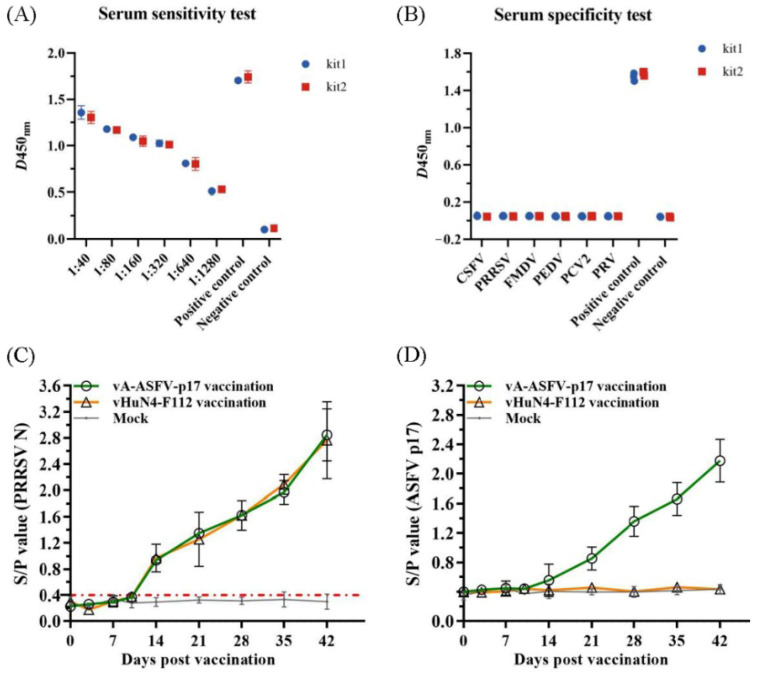
The indirect ELISA method against p17 efficiently detected clinical samples and recombinant PRRSV virus expressing ASFV antigen. (**A**) Sensitivity testing of the ELISA method using ASFV-positive serum. (**B**) Specificity testing of the ELISA method using CSFV, PRV, PRRSV, PEDV, FMDV, PCV2, and ASFV-positive sera samples. (**C**) The PRRSV-specific humoral immune response was assessed by the S/P value identified from serum samples collected at the indicated time points from piglets in vA-ASFV-p17, vHuN4-F112, and mock groups. (**D**) The ASFV-specific humoral immune response against p17 was tested using the indirect ELISA method from serum samples collected at the indicated time points from three groups.

**Table 1 viruses-15-00050-t001:** Primers used in this study.

Name	Sequences (5′-3′)
*P17*-F	CAGTGTGGTGGAATTCATGGACACAGAAACATCACCG
*P17*-R	GTGCTGGATATCTGCATCACTTTTCGAATTGTGGATGGGACCAAGAATGTGCCAGCTCCGCCA
pCold-TF-1-60-F	GGGTACCATGGACACAGAAACATCA
pCold-TF-1-60-R	GGAATTCTTAGTAGTACACGATGGCAAC
pCold-TF-50-117-F	GGGTACCATCTTGATTATCGTTGCC
pCold-TF-50-117-R	GGAATTCTTAAGAATGTGCCAGCTCCGC
pCold-TF-1-25-F	CGGGTACCATGGACACAGAAACATCA
pCold-TF-1-25-R	GGAATTCTTATCCTTGGGTGCTTTGCTT
pCold-TF-15-42-F	GGGTACCACTAGAGAAGGTATCAAG
pCold-TF-15-42-R	GGAATTCTTAAAGTAAAATAGCAGTGGT
pCold-TF-32-60-F	GGGTACCGCCAAGTACCCAGGCACC
pCold-TF-32-60-R	GGGAATTCTTAGTAGTACACGATGGCAAC

**Table 2 viruses-15-00050-t002:** Reference strains in this study.

NO.	Isolate	Country	Year	Accession No.	Genotype
1	ASFV SY18	China	2021	MH766894.1	II
2	ASFV Wuhan 2019-1	China	2019	MN393476.1	II
3	ASFV CADC_HN09	China	2019	MZ614662.1	II
4	ASFV Pig/HLJ/2018	China	2018	MK333180	II
5	ASFV China/2018/AnhuiXCGQ	China	2018	MK128995.1	II
6	ASFV SY-1	China	2022	OM161110.1	II
7	ASFV Belgium 2018	Belgium	2018	LR536725.1	II
8	ASFV Georgia 2007/1	Georgia	2007	FR682468.2	II
9	ASFV Tanzania/Rukwa/2017	Tanzania	2017	LR813622.1	II
10	ASFV CAS19-01-2019	China	2020	MN172368.1	II
11	ASFV NHV	Portugal	1968	NC_044943.1	I
12	ASFV LH60	Portugal	1960	NC_044941	I
13	ASFV E75	Spain	1975	NC_044958	I
14	ASFV Benin 97/1	Benin	1997	NC_044956	I
15	ASFV OURT	Portugal	1988	NC_044957	I
16	ASFV 25185_2008	Italy	2008	MW788410.1	I
17	ASFV30322	Italy	2013	MW736600.1	I
18	ASFV Nu1979	Italy	1979	MW723481.1	I
19	ASFV SD-DY-I-2021	China	2021	MZ945537.1	I
20	ASFV SPEC_57	South Africa	1985	MN394630.3	VIII
21	ASFV R35	Uganda	2018	MH025920.1	IX
22	ASFV Uvira B53	Congo	2021	MT956648.1	X
23	ASFV Zaire	Zaire	1977	MN630494	XX
24	ASFV RSA_2_2008	South Africa	2008	MN336500	XXII

**Table 3 viruses-15-00050-t003:** Tests of clinical samples.

Samples	Developed ELISA	Commercial ELISA
Positive	50	50
Negative	105	105

## Data Availability

The data that support the findings of this study are available from the corresponding author upon reasonable request.

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
