# Peer review of "The Indirect ELISA and Monoclonal Antibody against African Swine Fever Virus p17 Revealed Efficient Detection and Application Prospects"

_viruses, 2022, doi:10.3390/v15010050_

Round 1

Reviewer 1 Report

Li et al., produced a anti-p17 mAb with CHO-expressed p17 protein, identified a conservative B cell epitope and generated an indirect ELISA against ASFV p17. A recombinant PRRSV expressing p17 was rescued and the indirect ELISA efficiently monitored the p17 antibodies levels after recombinant PRRSV vaccination.

1. Line 158: vA-ASFV-p17 was passaged for 20 times. Was the p17 protein stably expressed during the 20 passages? And how about the recombinant virus virulence?

2. Line 62-63: p17 is detected as a specific antigen in the immunoreaction of pig sera with neutralizing antibodies. Did you detect the neuturalization effect of the pig sera obtained from the  vA-ASFV-p17-vaccinated pigs against ASFV? Could p17 potientially be used as vaccines?

Author Response

  1. Line 158: vA-ASFV-p17 was passaged for 20 times. Was the p17 protein stably expressed during the 20 passages? And how about the recombinant virus virulence?

Response: Thank you for your comments. Yes, by the serial cell passage of recombinant virus in vitro, IFA for vA-ASFV-p17 (P5, P10, P15 and P20 viral stocks) was performed. The results demonstrated that 6E3 specifically bound to vA-ASFV-p17-infected MARC-145 cells producing red fluorescence (Fig. 6I), indicating that p17 could be stably expressed in recombinant PRRSVs for 20 passages. Please check. The recombinant virus was rescued by reverse genetic manipulation, based on a PRRSV attenuated live vaccine strain vHuN4-F112. vHuN4-F112 is a commercially available HP-PRRSV vaccine and has reliable attenuated virulence to piglets.

  1. Line 62-63: p17 is detected as a specific antigen in the immunoreaction of pig sera with neutralizing antibodies. Did you detect the neuturalization effect of the pig sera obtained from the vA-ASFV-p17-vaccinated pigs against ASFV? Could p17 potientially be used as vaccines?

Response: Thank you for your comments. Your question is pertinent. I’m sorry that we didn’t do the neutralization test of ASFV by the sera of pigs immunized with recombinant virus vA-ASFV-p17. Because we don’t have BSL-3 laboratory or ABSL-3 animal facilities. We couldn’t operate live ASFV. Therefore, the neutralization effect of antibody was not directly evaluated. However, we have selected some recombinant viruses which were constructed and rescued in our lab to immunize pigs, which could provide good immune protection for oral challenge of ASFV (the experimental data have not been published). vA-ASFV-p17 is one of the components of the candidate vaccine combination. Meanwhile, we also detected the corresponding p17 antibody in immunized pigs. So, vA-ASFV-p17 can be potentially used as vaccine candidate. In this study, we aimed to obtain the detection method and mAb against p17, in order to get some useful tools for epidemiological surveillance and vaccine research on ASF.

Reviewer 2 Report

The authors expressed recombinant ASFV p17 protein using CHO cells and prepared a monoclonal antibody that successfully recognized p17. The epitope recognized by the monoclonal antibody was identified and was proved to be conservative among different strains. Using the recombinant p17 protein and the monoclonal antibody, they established an established ELISA method which has great potential for the specific and effective detection of clinical ASFV infection. Meanwhile, the established ELISA could detect the antibody produced by the recombinant PRRSV expressing p17. The present study provided a potential tool for epidemiological surveillance and vaccine development of ASF. There are some issues should be addressed to improve the quality of the study.

1.      ASFV p30 and p72 have been widely used for ASFV infection diagnosis. The rational for selection of p17 as the diagnostic marker of ASF should be provided in detail. What is the difference among the detection of p17, p30 and p72.

2.      Did the authors use the prokaryotic system for expression of ASFV p17 protein? Although CHO expressing system has more effective and precise post-translational modifications than that in the prokaryotic system. The prokaryotic system could also be used for expression of various viral proteins which could be used for diagnostic methods development.

3.      In Fig. 1D, why there were two bands of p17 could be detected using ASFV-positive serum?

4.      In Fig. 2, why the mAb subtypes were identified as both IgG2b for the heavy chain and kappa chain for the light chain?

5.      Did the 6E3 mAb recognize p17 protein in ASFV-infected cells or pigs?

6.      The information about using the recombinant PRRSV vector to express ASFV p30, p72 or other ASFV proteins should be discussed to provide insights for development of ASF vaccines.

Author Response

  1. ASFV p30 and p72 have been widely used for ASFV infection diagnosis. The rational for selection of p17 as the diagnostic marker of ASF should be provided in detail. What is the difference among the detection of p17, p30 and p72.

Response:Thank you for your comments. According to the coincidence rate experiments, 50 of 155 clinical serum samples tested positive by the established ELISA method, which was coincided with the result of the commercial kit. The commercial ELISA kit (ASFS-5P, Innovative Diagnostics, Grabels, France) was based on an indirect ELISA method against p30. The results indicated that the established ELISA method against p17 has same applicability with the commercial method against p30 among the current samples. The commercial ELISA kit against p72 was usually based on blocking ELISA, so we didn’t compare the difference. The reasons for selection of p17 as the diagnostic marker of ASF are as follows. Under the current prevailing situation, antibody detection of ASFV has become increasingly important. p30, p54, or p72 of ASFV have been the most researched and applied in clinical diagnosis. We think it is still necessary to explore more ASFV antigens. p17 is an abundant transmembrane protein localized at the viral internal envelope, which binds to the capsid protein p72 and closely connects the inner membrane and outer shell of ASFV virus. In particular, p17 is detected as a specific antigen in the immunoreaction of pig sera with neutralizing antibodies. Besides, we have obtained some recombinant PRRSVs in our lab to form the candidate vaccine combination. The detection method and mAb against p17 contributes to useful tools for epidemiological surveillance and vaccine research on ASF.

  1. Did the authors use the prokaryotic system for expression of ASFV p17 protein? Although CHO expressing system has more effective and precise post-translational modifications than that in the prokaryotic system. The prokaryotic system could also be used for expression of various viral proteins which could be used for diagnostic methods development.

Response:Thank you for your comments. We have tried to express various ASFV antigen proteins in prokaryotic system, but the effect is not ideal. Few proteins successfully expressed in the supernatants of E. coli BL21 cells. This would lead to complex purification process. ASFV p17 could not be expressed in our attempt. In this study, we also aimed to obtain the mAb against p17 in order to provide useful tools for vaccine research on ASF. Given the different post-translational modifications of CHO system and the above reasons, we finally chose mammalian cells.

  1. In Fig. 1D, why there were two bands of p17 could be detected using ASFV-positive serum?

Response:Thank you for your comments. There may be a small amount of miscellaneous protein in the purified p17, which could not be detected by SDS-PAGE. WB assay has higher sensitivity. Although the miscellaneous protein didn’t react with anti-strep tag, it showed another band using ASFV-positive serum. We thought it was non-target band of non-specific recognition.

  1. In Fig. 2, why the mAb subtypes were identified as both IgG2b for the heavy chain and kappa chain for the light chain?

Response:Thank you for your comments. According to the introduction of the mouse monoclonal antibody subtype identification kit (Proteintech), we identified the subtypes of 6E3. The prepared ascites diluted to 1:100,000, and then the sample was diluted to be tested appropriately and 50 μL/well was added to the sample well of the slat. Without incubation, 50 μL/well of 1× goat anti-mouse IgA+ IgM+ IgG-HRP was added to the sample wells. Next, the solution was mixed gently and the sides of the plate rack were lightly hand-tapped for 1 min and incubated at room temperature for 1 h. Next, the liquid in the hole was discarded, the plate washed 3 times and patted dry with absorbent paper. Then 100 μL of TMB was added to each well on the plate in the absence of light and at room temperature incubate for 10–20 min. Stop solution (100 μL/well) was added to each well. Finally, OD450nm was read with a microplate reader. The types of heavy chain included IgG1,IgG2a,IgG2b,IgG3,IgM,or IgA, and the types of light chain included Kappa or lambda. The results showed that the heavy chain of 6E3 was IgG2b, whereas the light chain was a kappa chain (Figure 2A).

  1. Did the 6E3 mAb recognize p17 protein in ASFV-infected cells or pigs?

Response: Thank you for your comments. I’m so sorry that we didn’t do the related experiments. Because we don’t have BSL-3 laboratory or ABSL-3 animal facilities and couldn’t operate live ASFV. We will seek the help of research institutions with BSL-3 laboratory to perform the corresponding research in future research.

  1. The information about using the recombinant PRRSV vector to express ASFV p30, p72 or other ASFV proteins should be discussed to provide insights for development of ASF vaccines.

Response: Thank you for your comments. We constructed a series of recombinant PRRSV expressing the major antigenic proteins of ASFV, including ASFV p30, p72, p17, p12, p54, pK205R etc. We had selected some recombinant viruses above to immunize piglets and did the immune efficacy against ASFV challenge. For each recombinant virus, the corresponding detection methods and antibodies are required. Immune efficacy and challenge tests are both necessary to find out the best combination. The experiments in our study will have a broad application prospect in development of novel ASF live vector vaccines.